# Natural Products in Polyclad Flatworms

**DOI:** 10.3390/md19020047

**Published:** 2021-01-21

**Authors:** Justin M. McNab, Jorge Rodríguez, Peter Karuso, Jane E. Williamson

**Affiliations:** 1Department of Biological Sciences, Macquarie University, Sydney, NSW 2109, Australia; justin.mcnab@hdr.mq.edu.au (J.M.M.); jorge.rodriguezmonter@australian.museum (J.R.); 2Department of Molecular Sciences, Macquarie University, Sydney, NSW 2109, Australia

**Keywords:** flatworms, Polycladida, tetrodotoxin, staurosporine, chemical ecology

## Abstract

Marine invertebrates are promising sources of novel bioactive secondary metabolites, and organisms like sponges, ascidians and nudibranchs are characterised by possessing potent defensive chemicals. Animals that possess chemical defences often advertise this fact with aposematic colouration that potential predators learn to avoid. One seemingly defenceless group that can present bright colouration patterns are flatworms of the order Polycladida. Although members of this group have typically been overlooked due to their solitary and benthic nature, recent studies have isolated the neurotoxin tetrodotoxin from these mesopredators. This review considers the potential of polyclads as potential sources of natural products and reviews what is known of the activity of the molecules found in these animals. Considering the ecology and diversity of polyclads, only a small number of species from both suborders of Polycladida, Acotylea and Cotylea have been investigated for natural products. As such, confirming assumptions as to which species are in any sense toxic or if the compounds they use are biosynthesised, accumulated from food or the product of symbiotic bacteria is difficult. However, further research into the group is suggested as these animals often display aposematic colouration and are known to prey on invertebrates rich in bioactive secondary metabolites.

## 1. Introduction

Animals in the marine environment host an assortment of secondary metabolites that are used for a variety of interactions. The majority have yet to be described and our understanding of the interplay of these compounds are slowly being investigated [1]. These molecules can have subtle or profound effects on biological interactions ranging from potent toxins that developed in a putative evolutionary arms races [2], to a range of antifouling metabolites influence larval settlement [3]. In a simplistic sense, natural products found in animals can be classified in three categories depending on their use: Defensive compounds, offensive compounds and signalling compounds. Marine compounds used for signalling are known as semiochemicals and can be produced by animals or sequestered from the environment and utilised for various purposes [4].

Semiochemicals are critical in inter- and intra-species communication in marine organisms. These bioactive molecules, when transported through a medium such as seawater, are detected by animals and act as signals to mediate their behaviour [4,5]. Animals, and especially invertebrates, have been investigated for semiochemicals that can be classified in either pheromones or allelochemicals, based on the transmission of information they relay [3,4,6,7,8]. Pheromones convey information about the emitter to a receiver of the same species. For example, the sea hares *Aplysia brasiliana* and *A. californica* are typically solitary animals but can form large mating aggregations that are coordinated by the discharge of three waterborne pheromones [9,10]. These pheromones show a sympatric effect on *Aplysia* and alone one will not attract an individual [10].

Allelochemicals can be further categorised by how they mediate interactions of emitters and receivers and include: Allomones, kairomones, apneumones and synomones. Allomones are used to benefit the emitter by inhibiting the growth and development of another receiver species [5]. Ascidians in the genus *Eudistoma* possess molecules that are toxic to several sessile invertebrate larvae and that prevent larvae from settling on the delicate exterior of the ascidian [11,12]. Kairomones convey beneficial information about the emitter to a receiver that is ultimately to the detriment of the emitter. They are colloquially known as “eavesdropping” chemicals because they are usually emitted as a by-product of the emitter and not intended for the receiver. For example, carnivorous polycerid nudibranchs such as *Robastra tigris* can locate their prey (*Tambja* spp.) via unique aldehydes in their slime trails [13,14]. Apneumones are signals from non-living materials that are utilised by a receiver to the detriment of an organism living on the material [15]. In marine systems there are no clear cases of molecules that could be considered apneumones. Synomones are used to benefit both the emitter and receiver of different species, such as the symbiont attraction and recognition between dinoflagellates and their hosts species [16,17]. In this example, dinoflagellates will excrete a signalling compound that allows the host to recognise and symbiotically sequester the cell [16]. The effects of semiochemicals are often subtle and in the marine ecosystem are generally understudied. Both defensive and offensive compounds are more comprehensively researched in marine systems as their effects are generally more obvious and profound.

Offensive and defensive molecules are directly associated with either assisting or preventing predation. These interactions are considered costly and metabolites are often more expensive to use, as such they are likely to be advertised by other signals like aposematic colouration or deimatic displays [18,19]. Marine invertebrates often adopt aposematic colouration when advertising chemical defence and one group that is well known for this are marine gastropods of the order Nudibranchia (nudibranchs), which are the subject of extensive natural products research [14,19,20,21,22].

A very similar, yet understudied order of animals are marine flatworms. Flatworms of the order Polycladida (referred to as polyclads herein) are generally considered to be toxic to predators despite a clear lack of testing for defensive or offensive compounds in most species [23,24,25]. This consideration is based on visual signals associated with polyclads as fish and nudibranchs share colouration and patterns with these flatworms suggesting Batesian or Mullerian mimicry [24,26,27]. Additionally, some juvenile fishes with shared colouration behave similarly to polyclads to enhance mimicry, suggesting that they are the Batesian mimics of polyclads [24,26]. Polyclad toxicity is advertised in aposematic colouration most commonly seen in the Cotylea suborder, and fish associate flatworm colouration with unpalatability [23]. However, the diversity of polyclad flatworms is large, and only a small number of eclectic species have been directly investigated for either toxic compounds or their ability to deter predators [23]. The first proof that polyclads were toxic was the pioneering work by Arndt in 1943, in which polyclad extracts were injected into isolated frog hearts and shown to produce cardiac arrest [28].

One compound that occurs within polyclad flatworms is the neurotoxin tetrodotoxin (TTX). Although TTX has only been documented in a few genera, mostly from Japan, the species that possess it are inconspicuous suggesting that it is not used in defence [29,30]. TTX illustrates the occurrence of bioactive metabolites in these polyclads and its distribution in the group is still being investigated. Although this highlights that species that display aposematic colouration could support this with chemical defences. This review synthesises knowledge firstly on polyclads and their ecology, then focuses on the diversity of molecules that have been isolated from polyclads to date, and then focuses on tetrodotoxin in a marine context. Suggestions for further research regarding natural products in flatworms are also given.

## 2. Results and Discussion

### 2.1. Overview of Polycladida

Platyhelminthes, commonly known as flatworms, are a group of bilaterian unsegmented invertebrates that are well established taxonomically [31]. The group is characterised by being acoelomate, dorsoventrally flattened, and possessing of a blind-sac intestine without an anus [31]. Organisms belonging to this group can be free-living, however, this group is most classically known for its parasitic members [31]. Flatworms from the order Polycladida are a group of free-living (non-parasitic) marine worms found in a range of habitats including coral reefs, rocky shores, soft sediments and deep-sea environments [23,24]. Polyclads possess a plicate pharynx that they use to prey on other small invertebrates such as molluscs or crustaceans. The feeding method usually involves the eversion of the pharynx and swallowing of the prey whole, with any unassimilated remains being discharged to the exterior of the mouth. These animals have minor amounts of cephalisation with a central ganglion connected by a ladder nervous system [31]. They are relatively limited when it comes to sensing their environment, only possessing basic chemosensory receptors allowing the detection of changes in water flow [32]. Polyclads also possess simple light sensitive ocelli or “eyespots” that detect the presence or absence of light, but are unable to process images [24,31]. These eyespots are one of the few external morphological traits used in species identification and come in four different arrangements depending on their position: Cerebral, tentacular, marginal and frontal [31].

Polyclads are divided into two suborders: Acotylea and Cotylea. The apomorphy that separates the two groups is the presence of a ventral sucker (cotyl) located in the posterior end of their body, with acotyleans lacking a ventral sucker and cotyleans possessing a ventral sucker behind the female genital opening [24,31]. Polyclads from the suborder Cotylea are more commonly found subtidally inhabiting coral reefs and displaying conspicuous colouration patterns, while those of the suborder Acotylea are usually found hiding beneath boulders in the intertidal region and possess inconspicuous colouration. All polyclad flatworms are hermaphroditic, with the male and female genital systems occurring sequentially, usually in the posterior region of the body [31]. Due to their relatively simple anatomies, polyclad species are mainly identified by the reconstruction of the internal anatomy of their reproductive systems, as variations of these occur between closely related species enabling a clear distinction [24,33]. In present times, molecular data are used alongside morphology as another source of evidence for species identification [34,35].

### 2.2. Ecology of Polyclads

#### 2.2.1. The Roles of Flatworms in Their Environments

Within the marine ecosystem, polyclads typically prey on in the communities they inhabit. All polyclads are carnivorous and act as mesopredators of smaller invertebrates such as crustaceans, ascidians, gastropods, and shellfish [36,37,38,39]. The specific diets of polyclads are not always known, however, some species have been observed to overcome the secondary metabolites of their prey and may act as specialised hunters [36,37,40]. Community impacts of flatworms remain undocumented, but this group has the potential to affect populations of small invertebrates in communities they inhabit, particularly intertidal rock pools and benthic systems [38,41]. In some areas flatworms have been shown to impact the demography of sessile species such as barnacles but only in the absence of higher trophic levels, suggesting that their influence on their community is dynamic and complex [41].

#### 2.2.2. Commercial Pests

The most studied cases of predation by polyclads are in relation to the damage they cause to the commercial shellfish industry, and to a lesser extent the coral aquaculture industry [42,43,44,45,46,47]. Globally, shellfish stocks such as oysters (pearl and food), mussels and clams suffer predation by flatworms, with members of the family Stylochoidea typically responsible for the majority of damage [48,49]. The polyclad *Stylochus* (previously *Imogine*) *mcgrathi* can consume up to 4.9 mg of oyster flesh per day, depending on size [42] and current methods of control predation on commercial oyster stocks involve time consuming treatments such as regular hypo- and hypersaline baths [42]. Despite the well documented cases of polyclads predating on shellfish globally [43,44,49], how the flatworm enters the shell of its prey remains controversial [45]. Discovering ways to control flatworms is of high importance to aquaculture and wild harvest industries and could result in substantial financial savings [43].

#### 2.2.3. Symbiosis

Beyond their role as predators, some polyclads live in intimate association with other invertebrates (e.g., corals, hermit crabs, echinoderms and molluscs), forming a variety of symbiotic relationships [24]. The association between a polyclad and its host is purely commensal and cases of true parasitism involving polyclads have not been established, despite being well documented in the closely related acetol flatworms [32,50]. Bo, et al. [51], for example, gives evidence of highly cryptic flatworms that are specific symbionts to black corals (Order Antipatharia) in the Mediterranean Sea. The authors suggest a symbiotic relationship exists between these organisms as there appears to be clear co-evolution between the species. Colouration from the flatworm matches that of the coral, and there are striking similarities between the flatworm egg morphology and the nodules of the coral polyp [51]. This relationship is inferred, however, as polyclads do not contain spiracles in their digestive tract, suggesting a lack of predation between the two animals [51]. As black corals are grazed by mesopredators, such as ovulids, copepods and amphipods [51,52] that are preyed upon by flatworms [53], a parsimonious suggestion is that the flatworms remove the mesopredators from the corals and thus provide benefit that supports symbiosis.

Polyclads are also reported as being endosymbiotic to echinoderms [54,55] and some molluscs [56,57,58]. In the instances of polyclads residing within the mantle cavities of molluscs, the relationship is more one of commensalism than of true symbiosis [57]. In this occasion, the coastal polyclad *Comoplana pusilla* (previously *Stylochoplana*) in Japan seeks refuge within the shells of snails to avoid desiccation, the polyclads themselves do not benefit the snails [57,58].

### 2.3. Chemical Ecology of Polyclads

Few secondary metabolites have been isolated from polyclads and the order remains mostly a prospect for natural products research. However, their varied diets and aposematic coloration suggest the presences of toxic molecules associated with defence [37,59]. Investigated polyclads are in both suborders Acotylea and Cotylea and include species from the genera: *Pseudoceros* and *Prostheceraeus* from Cotylea; and *Planocera*, *Stylochoplana*, *Stylochus*, *Notoplana* and *Notocomplana* from Acotylea (Table 1).

#### 2.3.1. Staurosporine

Including staurosporine (Figure 1, **1**), ten other analogues (**2**–**11**) were also isolated and confirmed by NMR spectroscopy from *Pseudoceros* sp. (later described as *P. indicus*), of these, eight were new and found in both the polyclad and its food, the ascidian *Eudistoma toealensis* [59,60]. Staurosporine is characterised by an indolocarbazole structure that is a non-specific inhibitor of protein kinases [61]. These molecules block the binding of ATP to the catalytic domain in >95% of kinases and effectively prevents phosphorylation [61,62]. Staurosporine is accepted as a bacterial metabolite synthesised by Gram-positive bacteria of the genus *Actinomycetes* which can be symbiotic in the ascidian, this then allows the host to defend itself from predation due to its high cytotoxicity [61,62,63]. These compounds have a limited distribution and are typically confined to bacteria, ascidians and a prosobranch gastropod [64,65]. The notion that polyclads can sequester this molecule from their prey implies an adaptation to prevent kinase inhibition. This molecule would be valuable to the flatworms, however, the species that contain staurosporine are yet to be tested for defensive capabilities. Other bioactive natural products have been isolated from these ascidians but not from the polyclads, including pentachlorooctatriene, and this suggests that secondary metabolites are selectively sequestered by polyclads [66].

#### 2.3.2. Pseudocerosine

The alkaloid pseudocerosine (**12**) was isolated by Hemscheidt as a distinct blue pigment located around the margin of *P. indicus* [67]. It was originally incorrectly described as an azepinoindole molecule; however, a recent revision of its structure by Sperry identified it as a pyridoacridine, but a new branch of the pyridoacridine family tree [68]. Pyridoacridines are a unique group of purely marine alkaloids that are often cytotoxic. They have been isolated from sponges, molluscs, tunicates, and cnidarians and it is suspected that **12** is trophically sequestered from the polyclad’s prey [69,70]. It is unclear how **12** contributes to the defence of the polyclad considering several staurosporines were also isolated from this species [67,71]. Sperry reported a failed attempt to synthesise the incorrect structure and then subsequently synthesised the correct structure to show that its spectral data matched that of the natural product [71]. The corrected synthesis was done in 8 steps and created two diastereomers of **12** in *syn* and *anti-*configuration, with yields of 30% for the former and 43% for the latter (see supplementary material of [68]). The biological activity of **12** is reported against only human adenocarcinoma (SKOV-3) cancer cells where it showed mild cytotoxic activity (25 µg/mL) [67].

#### 2.3.3. Villatamine

The North Sea cotylean flatworm *Prostheceraeus vittatus* contains one novel group of molecules, the villatamines (**13**,**14**) as well as three lepadins (**15**–**17**), a group that had previously been isolated from its preferred prey species, the ascidian *Clavelina lepadiformis* [40]. Villatamines (**13**,**14**) are a long-chain pyrrolidine alkaloids from *P. vittatus*, from where the name is derived [40,72]. In their pure form, villatamines are relatively unstable oils that occur at concentrations of 0.02 mg/individual for villatamine A (**13**), and 0.09 mg/individual for villatamine B (**14**) [40,72]. There is not much research on these molecules but their synthesis is described in Hu, et al. [73] with a 26% overall yield after six steps for **13**, and 37% yield after five steps for **14**. Villatamine B (**14**) showed cytotoxic activity to several forms of cancer cells, including, murine leukaemia (P388, ED_50_ 11.4 µg/mL), and human breast cancer (MCF7, 2.8 µg/mL), glioblastoma/astrocytoma (U373, 1,9 µg/mL), ovarian carcinoma (HEY, 2.8 µg/mL), colon (LOVO 1.7 µg/mL) and lung (A549, 2.8 µg/mL) cancer cells [40,73]. In an ecological sense the cytotoxic activity observed in human cells is suspected to translate to that of a feeding deterrent in other organisms, emulating sequestered compounds in nudibranchs [40].

#### 2.3.4. Lepadins

Lepadins (**15**–**17**) are decahydroquinoline alkaloids that have been isolated from the polyclad *P. vittatus* and its prey [40,66]. Lepadin A (**15**) was previously isolated from *C. lepadiformis* by Steffan [66]. Lepadins sequestered by *P. vittatus* occur at different concentrations; **15** was found to occur at 0.2 mg/individual, **16** was occurring at 0.04 mg/individual and **17** at 0.02 mg/individual [40]. These molecules similarly show cytotoxic activity to an identical range of cancer cells including murine leukaemia (**15**: 1.2 µg/mL, **16**: 2.7 µg/mL), human breast (**15**: 2.3 µg/mL, **16**:,17 µg/mL), glioblastoma/astrocytoma (**15**: 3.7 µg/mL, **16**: 10 µg/mL), ovarian carcinoma (**15**: 2.6 µg/mL, **16**: 15 µg/mL), and colon (**15**: 1.1 µg/mL, **16**: 7.5 µg/mL) and lung (**15**: 0.84 µg/mL, **16**: 5.2 µg/mL) cancer cells [40]. Lepadins have additionally been demonstrated to inhibit nicotinic acetylcholine receptors in the nervous system [40,74]. In a more ecological sense, **15** has previously been illustrated to be profoundly toxic to a range of different larvae, including, tubeworm, bryozoan and hydroid larvae [12]. These results are inferred to represent toxicity against predators and lepadins are expected to play a role in defence within *P. vittatus* [12].

#### 2.3.5. Tetrodotoxin

Since the discovery of tetrodotoxin (TTX, **18**) in polyclads in 1986, the presence of TTX has been described in five polyclad genera in the Acotylea suborder. Unfortunately, concentrations are only reported for two genera (Table 1) [29,30,75,76,77]. The first occurrence of TTX in flatworms was documented in Japan in 1986, where a study classified individuals in loose groups based on body types, then tested individuals for TTX [75]. Individuals tested contained a wide range of TTX from between <2 mouse units per gram (1 MU/g = a lethal dose for one mouse) to dangerous levels of 1508 MU/g. The lack of taxonomic distinction of species could mean that tested polyclads encompass species from both suborders.

The most recent evaluation of TTX in flatworms focused on the Acotylea suborder and tested 12 species of polyclads from the Kanagawa prefecture, Japan [30]. In this study, only two species of *Planocera* contained TTX, whist species previously described as toxic, including *Stylochus* and *Notoplana*, were free of TTX [30]. This paper tested a wide range of genera to obtain a more representative measure of Acotylea toxicity and found that TTX occurs exclusively within the Planoceridae family [30]. Additionally, the authors suggest that individuals of another genera, *Stylochoplana*, which had been confirmed to contain TTX were misclassified, and actually belong to *Planocera* [30]. This is understandable as polyclads are classified on internal morphology and DNA samples were not taken [77]. As such, the most recent evidence suggests that in Acotylea TTX is confined to flatworms in Planoceridae (Table 1). This discovery is important as it has been observed that pufferfish selectively consume these polyclads presumably to toxify themselves with TTX [76,78,79,80]. This may be the primary natural source of tetrodotoxin in pufferfish which are known to become non-toxic when net cultured and fed a non-toxic diet [81]. In this genus of flatworms, **18** and nine analogues (**19**–**27**) have been isolated from four different species (Table 1) [30,82]. The presence of this many analogues in flatworms resulted in Yotsu-Yamashita, Abe, Kudo, Ritson-Williams, Paul, Konoki, Cho, Adachi, Imazu and Nishikawa [82] inferring that several analogues could be used in TTX biosynthesis in animals as the analogues act as precursors [82]. The process is thought to start with **27** which is then oxidised in two intermediate steps into **18**, the origin of the precursor molecule is only hypothesised as being provided by microorganisms Yotsu-Yamashita, Abe, Kudo, Ritson-Williams, Paul, Konoki, Cho, Adachi, Imazu and Nishikawa [82].

In planocerid flatworms, TTX is suggested to assist in prey capture and in defence of their egg masses as the largest concentrations of the toxin are found in the pharynx, eggs and ovaries [53,81,82]. Prey capture is reported to be greatly enhanced by TTX as stated in Ritson-Williams, Yotsu-Yamashita and Paul [53], who noted a large range of often mobile invertebrates that were preyed on by the tested flatworm. A puzzling feature of TTX in flatworms is the lack of evidence of specific housing cells that might assist in envenomation or movement of the toxin [53,80,83]. Such specialised cells or venom-delivery systems are common in organisms known for harnessing secondary metabolites, as they reduce any opportunity of autotoxicity and maximise the effectiveness of toxin [84]. This distinction is important as previous work has recognised that TTX is able to inhibit neural cell depolarisation in polyclads, and if TTX encounters neural cells then the animal could be negatively affected [85]. Other animals like octopi that house TTX for use as a venom typically show the greatest concentration of TTX in salivary glands, which have specialised morphology that gives them greater control of the venom as it is ecologically or metabolically costly to use [86,87,88]. As such, further physiological studies with respect to TTX usage in flatworms are warranted. Additionally, understanding the ubiquity of TTX within polyclads requires assessment of species from both Acotylea and Cotylea.

### 2.4. Overview of Tetrodotoxin

Tetrodotoxin is a tricyclic guanidinium-based neurotoxin, it is heat stable and also water soluble, making it relatively stable within the marine environment [90]. Tetrodotoxin works by binding to voltage-gated sodium channels in neural and muscular cells (*K*_D_ = 10^−10^ M, Figure 2). This is achieved by binding to the sodium binding site, essentially blocking the channel, and forcing the intracellular gate closed, rendering it unable to depolarise [90,91,92]. Specifically, TTX will bind in the SS2 region of sodium channels. In this region the hydroxyls at C-6, C-9, C-10 and C-11 are essential for tetrodotoxin to bind with a high binding affinity [93,94]. The effects of this binding inhibits the release of neurotransmitters from the affected cell and leads to eventual paralysis as cells are unable to depolarise [90]. Organisms that produce TTX also typically produce a variety of analogues of TTX, each with differing toxicities due to different types of small alterations of the chemical structure that affect the binding affinities of the toxin to sodium channels in different species [85,88].

TTX has been adapted for different uses and as such is dispersed differently within the animals that utilise it; however, similarities are seen based on usage and method of sourcing TTX. Common locations of TTX storage include the mantle, the ovaries of animals that use TTX defensively [95,96,97], the salivary or mandibular glands, or mouthparts of animals that use it offensively [53,87,98]. Additionally, for animals that acquire TTX from dietary sources, high concentrations of the toxin and its analogues have been observed in detoxifying organs such as the gastrointestinal tracts, liver/nephridia and sometimes the gills [29,88,99]. Animals that contain TTX are also widely distributed throughout the world; with species of toxic pufferfish occurring throughout Asia [29,97,100], and toxic invertebrates occurring within the Oceania region [87,101,102] as well as from Europe [102,103,104,105]. This wide distribution of TTX in the marine environment suggests a more important ecological role for this toxin than currently recognised.

Animals that contain TTX maintain different ecological roles; however, testing for ecological significance is not standard practice when studying this toxin. Animals that possess TTX span several trophic levels from higher order predators like pufferfish or octopi to primary producers [2,98,106]. This toxin is observed in a widespread distribution of animals from different niches, including detritovores and filter-feeders, parasites, predators, and omnivores [88,102,107,108]. With such a broad phylogenetic spread of animals utilising TTX, the bioaccumulation pathways that this molecule travels from primary producers to predators are difficult to track. To date, pufferfish, gobies and nudibranch are the only examples of animals predating on other species specifically for the purpose of toxifying themselves with TTX exist [77,80,97].

## 3. Conclusions

Marine natural products research has knowledge gaps especially in the benthic communities and invertebrates such as polyclad flatworms. These are often overlooked because reliable sources of individuals of the same species are hard to come by. However, due to the diverse range of environments they inhabit, their often-aposematic colouration and the fact that novel natural products have recently been isolated from them, these animals could be host for more compounds and deserve further research. In the context of TTX, our understanding of which species contain the toxin, and which do not is still unclear. For those that do contain TTX the ecological relevance of the activity of the natural product to the species also needs to be clarified.

Tetrodotoxin is suggested to act as a keystone chemical as it occurs in a range of phyla, shows potent effects with small concentrations in animals it infects, and as such causes cascading changes to the ecology of systems it inhabits [109]. This effect is most evident in freshwater riparian ecosystems but is yet to be illustrated in a marine context [110]. Additionally, TTX has several analogues that differ structurally from TTX and as such, affect sodium channels differently. The cause of this difference could be to maintain this toxin’s effectiveness against adapted sodium channels or ion channel from many different species. Another molecule that we suggest is a prospective keystone chemical is staurosporine for similar reasons. Staurosporine is found in unrelated phyla including ascidians, flatworms, and molluscs [59,65]. This molecule and its analogues are pan-kinase inhibitors and cytotoxic to cancer cells (monocyclic leukaemia MONO-MAC6, IC_50_ 13.3 ng/mL), which translates to a high toxicity against ecologically relevant sources such as fish and crustaceans [65,111,112,113].

To classify TTX and staurosporine as a keystone chemical, the ecological consequences of this molecules’ occurrence must be investigated. Namely, how animals that possess these molecules deter their predators, and what the cascading ecological effects from those interactions are. This could be done through better describing predator prey interactions, investigating the range of organisms that possess these molecules in an ecosystem and by comparing biological communities to a control site.

Of the roughly 800 described species of polyclads [114] only about 12 species have been tested for natural products (Table 1) with some only being investigated for TTX (see [30] for details), from those, 27 bioactive metabolites have been isolated, with 15 of those being novel. The 15 novel compounds all originated from flatworms of the Cotylean suborder, and as these species are typically aposematic, it is highly likely that other species contain novel compounds. To this end, we suggest further studies be conducted on the natural products of Cotylean flatworms. Studying this group will likely not only reveal unique molecules, but it will also clarify if polyclads are Batesian or Mullerian mimics of the nudibranchs and fish that they share aposematic colours with.

Several of the molecules isolated from polyclads, such as staurosporine and its analogues (**1**–**11**) and TTX and its analogues (**18**–**27**), are known to be sourced from bacteria, including *Streptomyces*, *Actinomycetes*, *Pseudomonas* and *Vibrio* [62,115]. For both these groups of molecules, the compound is accumulated by the animal from the bacteria that produces it, in the case of staurosporine, ascidians that consume the bacteria or have it as a symbiont, are the source that polyclads use to obtain the toxin [59,60]. In the case of TTX and analogues, the source of the toxin in polyclads is unclear, these molecules are known to trophically accumulate to animals, but symbiotic relationships between bacteria and animals with TTX are also common [115]. As such it is possible that flatworms could either be accumulating TTX from their diet or from symbiotic relationships with bacteria. Discovering where polyclads acquire their toxins from could give a clearer understanding of the ecology of these molecules and their influences on the biological communities. As such we suggest sequencing the gut microbiome of polyclads to understand the sources of their toxins.

## Figures and Tables

**Figure 1 marinedrugs-19-00047-f001:**
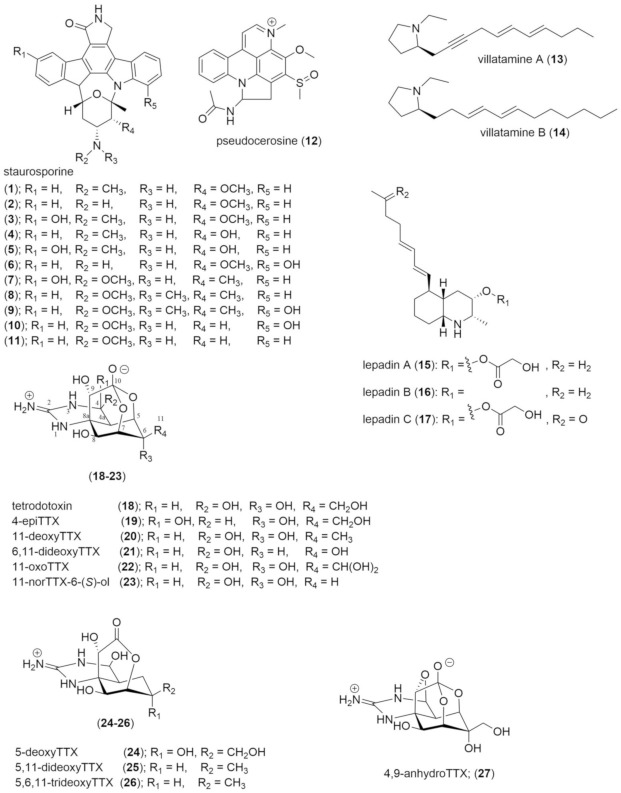
Structures of compounds isolated from the Order Polycladida.

**Figure 2 marinedrugs-19-00047-f002:**
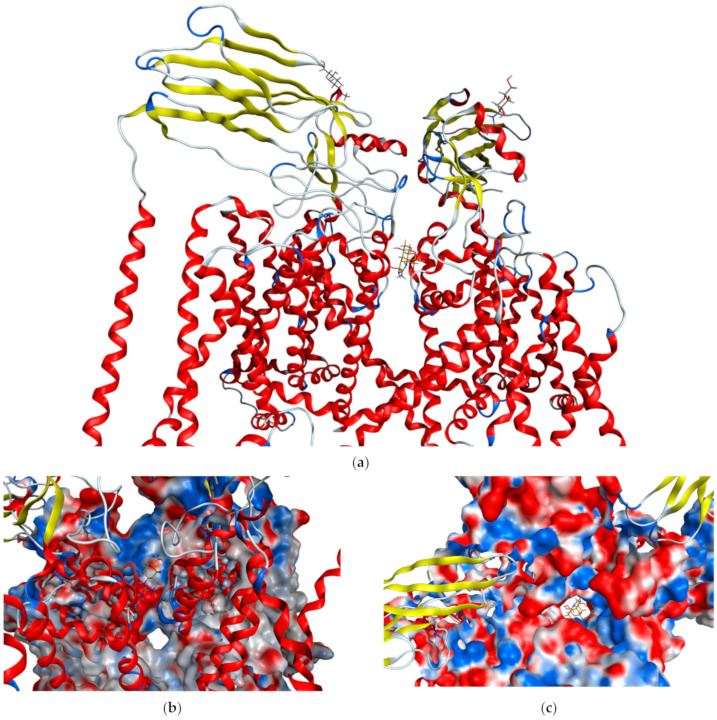
Crystal structure of human Nav1.7 sodium channel in complex with auxiliary subunits and tetrodotoxin (gold) [92]. (**a**) Side view showing transmembrane helices in red and axillary proteins in yellow. TTX is in the centre of the ion channel. (**b**) Side view showing TTX (gold) lodged in the top of the ion channel and (**c**) Top view showing molecular surface with TTX (gold) blocking the ion channel.

**Table 1 marinedrugs-19-00047-t001:** Natural products from flatworms of the Order Polycladia.

Sub-Order	Family	Species	Compound	Location	conc. (mg/g)	Geography
Acotylea	Planoceridae	*Planocera* cf *heda* [79]	**18**	Whole body tested	249 (n = 1) + 1351 (n = 1)	Japan
		*Planocera multitentaculata* [30,89]	**18**	Testis + Pharynx + Ovaries + Eggs	93 ± 50	Japan
			**18**	Testis + Pharynx + Ovaries + Eggs	272 ± 508	Japan
		*Planocera* sp. [53,82]	**18**–**27**	Pharynx + Gastro-intestinal tract + Mantle	Not reported	Guam
	Stylochoplanidae	*Stylochoplana* sp. [77]	**18**	Eggs	108,000 ± 2000	New Zealand
		*Stylochoplana clara* [29]	**18**	Not mentioned	Not reported	Japan
	Stylochoidea	*Stylochus orientalis* [29]	**19**, **24**	Whole body tested	22–50.6	Taiwan
		*Stylochus ijimai* [29]	**18**	Not mentioned	Not reported	Japan
	Notoplanidae	*Notoplana humilis* [29]	**18**	Not mentioned	Not reported	Japan
	Notopcomplanidae	*Notocomplana koreana* [29]	**18**	Not mentioned	Not reported	Japan
Cotylea	Pseudocerotidae	*Pseudoceros indicus* [59,60]	**1**–**12**	Whole body tested	Not reported	Micronesia
		*Pseudoceros tristriatus* [59]	**1** (presumed)	Not mentioned	Not reported	Micronesia
	Euryleptidae	*Prostheceraeus vittatus* [40]	**13**–**17**	Whole body tested	Not reported	North Sea

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
