# Peer review of "Natural Products in Polyclad Flatworms"

_marinedrugs, 2021, doi:10.3390/md19020047_

Round 1
Reviewer 1 Report
For a long time, TTX--intoxication mechanism in pufferfish had been unknown, We suggested that TTX-intoxication mechanism in pufferfish comes from a food-chain by predation of TTX bearing organism and confirmed that non-toxic pufferfish would be produced by net-culture giving non-toxic diet in sea or land room in order to prevent from their TTX intoxication. If without the above net-culture, pufferfish will try to seek for TTX containing organisms which is natural phenomena, and become toxic with TTX. Our suggestion was right because we can succeed to produce non-toxic pufferfish by the above net-culture.Author Response
Reviewer 1
For a long time, TTX--intoxication mechanism in pufferfish had been unknown, We suggested that TTX-intoxication mechanism in pufferfish comes from a food-chain by predation of TTX bearing organism and confirmed that non-toxic pufferfish would be produced by net-culture giving non-toxic diet in sea or land room in order to prevent from their TTX intoxication. If without the above net-culture, pufferfish will try to seek for TTX containing organisms which is natural phenomena, and become toxic with TTX. Our suggestion was right because we can succeed to produce non-toxic pufferfish by the above net-culture.
We thank the author for his insights and have added a sentence to emphasise the point that non-toxic puffer fish could be cultured in the absence of toxic prey.
The sentence below has been added to lines: 285-286 after the first mention of pufferfish in the text:
“This may be the primary natural source of tetrodotoxin in pufferfish which are known to become non-toxic when net cultured and fed a non-toxic diet[82]”
Reviewer 2 Report
This is a nice review of natural products found in polyclads, a class of turbellarian worms. By bringing together current data on this group it will stimulate further research.
One source of information was overlooked, perhaps because it was originally published in German language, namely the pioneering research of Arndt. Several well known books (in English) on animal toxins can be consulted for his findings, including Halsteads Poisonous and Venomous Marine Animals of the World, and Baslow's Marine Pharmacology. I believe Hyman's volume (already cited) also mentions Arndt's papers.
The authors use an unusual word, "predate" instead of "prey"; when predate was found in Websters Dictionary it was defined as "to antedate", which has a completely different meaning. It is strongly recommended that prey be used rather than predate.
Author Response
Reviewer 2
This is a nice review of natural products found in polyclads, a class of turbellarian worms. By bringing together current data on this group it will stimulate further research.
Thank you for the acknowledgment.
One source of information was overlooked, perhaps because it was originally published in German language, namely the pioneering research of Arndt. Several well known books (in English) on animal toxins can be consulted for his findings, including Halsteads Poisonous and Venomous Marine Animals of the World, and Baslow's Marine Pharmacology. I believe Hyman's volume (already cited) also mentions Arndt's papers.
We have now added citations to Arndt’s pioneering work. The changes “in most species” has been made to lines 75-76, and the sentence “The first proof that polyclads were toxic was the pioneering work by Arndt in 1943, in which polyclad extracts were injected into isolated frog hearts and shown to produce cardiac arrest [28]”. has been added to lines 83-85. Reference 28 is Arndt 1943, “Polyclad und maricole tricladen als giftträger”.
The authors use an unusual word, "predate" instead of "prey"; when predate was found in Websters Dictionary it was defined as "to antedate", which has a completely different meaning. It is strongly recommended that prey be used rather than predate.
We agree that Predate is virtually never used in spoken English, even though it is correct and relates to predator. Prey on is more frequently used so we have changed all occurrences. Also, because the word pre-date, has another meaning, we have decided to change predate to “prey on” throughout:
Changes were made on line 20, 116, 143 and 181 (181 was changed from predated on, to preyed upon).